# A Cysteine Residue within the Kinase Domain of Zap70 Regulates Lck Activity and Proximal TCR Signaling

**DOI:** 10.3390/cells11172723

**Published:** 2022-09-01

**Authors:** Annika Schultz, Marvin Schnurra, Ali El-Bizri, Nadine M. Woessner, Sara Hartmann, Roland Hartig, Susana Minguet, Burkhart Schraven, Luca Simeoni

**Affiliations:** 1Institute of Molecular and Clinical Immunology, Medical Faculty, Otto-von-Guericke University, 39120 Magdeburg, Germany; 2Faculty of Biology, Signalling Research Centres BIOSS and CIBSS, University of Freiburg, 79085 Freiburg, Germany; 3Spemann Graduate School of Biology and Medicine (SGBM), University of Freiburg, 79085 Freiburg, Germany; 4Center of Chronic Immunodeficiency CCI, University Clinics and Medical Faculty, 79106 Freiburg, Germany; 5Health Campus Immunology, Infectiology and Inflammation (GC-I3), Medical Faculty, Otto-von-Guericke University, 39120 Magdeburg, Germany; 6Center for Health and Medical Prevention (CHaMP), Otto-von-Guericke-University, 39120 Magdeburg, Germany

**Keywords:** Zap70, TCR-ζ, TCR signaling, C564, Lck, T-cell activation, signal propagation, LAT signalosome

## Abstract

Alterations in both the expression and function of the non-receptor tyrosine kinase Zap70 are associated with numerous human diseases including immunodeficiency, autoimmunity, and leukemia. Zap70 propagates the TCR signal by phosphorylating two important adaptor molecules, LAT and SLP76, which orchestrate the assembly of the signaling complex, leading to the activation of PLCγ1 and further downstream pathways. These events are crucial to drive T-cell development and T-cell activation. Recently, it has been proposed that C564, located in the kinase domain of Zap70, is palmitoylated. A non-palmitoylable C564R Zap70 mutant, which has been reported in a patient suffering from immunodeficiency, is incapable of propagating TCR signaling and activating T cells. The lack of palmitoylation was suggested as the cause of this human disease. Here, we confirm that Zap70^C564R^ is signaling defective, but surprisingly, the defective Zap70 function does not appear to be due to a loss in palmitoylation. We engineered a C564A mutant of Zap70 which, similarly to Zap70^C564R^, is non-palmitoylatable. However, this mutant was capable of propagating TCR signaling. Moreover, Zap70^C564A^ enhanced the activity of Lck and increased its proximity to the TCR. Accordingly, Zap70-deficient P116 T cells expressing Zap70^C564A^ displayed the hyperphosphorylation of TCR-ζ and Zap70 (Y319), two well-known Lck substrates. Collectively, these data indicate that C564 is important for the regulation of Lck activity and proximal TCR signaling, but not for the palmitoylation of Zap70.

## 1. Introduction

Upon antigen recognition, protein tyrosine kinases belonging to the Src and Syk families initiate TCR signaling cascades, leading to T-cell activation (for a review, please see [1]). The Src family kinase Lck is the first kinase to be activated. Lck, in turn, phosphorylates two tyrosine residues located within the immunoreceptor tyrosine-based activation motifs (ITAMs) of the ζ-chain in the TCR/CD3 complex. Doubly phosphorylated ITAMs function as high-affinity binding sites for the tandem SH2 domains of a second tyrosine kinase, Zap70, belonging to the Syk family [2]. The binding of Zap70 to the ITAMs allows for its recruitment to the TCR/CD3 complex and, in addition, destabilizes the inactive/closed conformation of Zap70. Subsequently, Zap70 is phosphorylated by Lck on Y319, located in the interdomain B, which results in the activation of Zap70. The full activation of Zap-70 also depends on the auto- trans-phosphorylation of Y493, located in the kinase domain [3]. The importance of Zap70 in signal propagation has been additionally demonstrated by the fact that Zap70-deficient T cells display an impaired phosphorylation of LAT and SLP76 [4,5], two Zap70 substrates, which compromises the assembly of the LAT signalosome and impairs the activation of downstream signaling cascades [4,6]. In agreement with the important role of Zap70 in TCR signaling, Zap70^−/−^ mice exhibited a severe block at the DP stage of thymic development, correlating with a defective positive selection [7,8].

In humans, Zap70 has been extensively studied for its involvement in many pathologies including B-cell-derived malignancies [2,9], autoimmunity [10], and severe combined immunodeficiencies (SCID) (recently reviewed in [2,11]). Most of the Zap70 mutations found in SCID patients affect the kinase domain [11]. One particular Zap70 mutation, C564R, resulted in a form of SCID characterized by recurrent pneumonia, oral candidiasis, exfoliative dermatitis, and subcutaneous nodules [12]. A recent study has investigated the molecular mechanisms underlying this form of SCID [13]. The authors of this study generated a Zap70^C564R^ mutant that was stably expressed in the Zap70-deficient Jurkat T-cell line P116. The data show that Zap70^C564R^ (i) retains its enzymatic activity, (ii) is recruited to the plasma membrane upon TCR stimulation, (iii) is hyperphosphorylated on Y319 and Y493, and (iv) displays an increased association with Lck. Nevertheless, despite retaining its enzymatic activity, Zap70^C564R^ failed to propagate the signal, as suggested by the impaired phosphorylation of its substrates LAT and SLP76. Thus, T cells expressing the C564R mutation failed to activate downstream pathways and to induce T-cell activation. Additional mechanistic analyses revealed that C564 is S-acylated, an event that has been proposed as being required to bring Zap70 close to its substrates LAT and SLP76 [13]. Collectively, these data propose a novel mechanism regulating T-cell activation and explain at the molecular level the cause of the immunodeficiency induced by the C564R mutation.

In this study, we generated a C564A mutant to shed light on the mechanisms regulating Zap70 function. Conversely to the study of Tewari and co-workers [13], we surprisingly found that Zap70^C564A^ was not only signaling-competent, but also capable of activating downstream pathways. As a C-to-A mutation abrogates S-acylation, our data demonstrate that the S-acylation of Zap70 on C564 is not required for signal propagation. Moreover, we have found that Zap70^C564A^ enhances the activation of Lck and increases its proximity to the TCR. Accordingly, TCR-ζ and Y319 of Zap70, two well-known Lck substrates, are hyperphosphorylated in cells expressing Zap70^C564A^. Thus, C564 of Zap70 is involved in the regulation of Lck activity and proximal TCR signaling, independent of the S-acylation.

## 2. Materials and Methods

### 2.1. Antibodies

The antibodies depicted in Table 1, Table 2 and Table 3 were purchased from BD Pharmingen^TM^ (San Diego, CA, USA), Dianova (Hamburg, Germany), Sigma-Aldrich (Saint Louis, MO, USA), Everest Biotech Ltd (Bicester, UK), Cell Signaling Technology (Danvers, MA, USA), Merck Millipore, Upstate (Burlington, MA, USA) and Santa Cruz Biotechnology (Dallas, TX, USA).

### 2.2. Cell Culture

Zap70-deficient P116 T cells and Lck-deficient J.Lck [14] were grown in RPMI 1640 (Roswell Park Memorial Institute) medium with 10% FBS fFetal bovine serum) as well as 1% penicillin/streptomycin. HEK293T-TCR [3] and HEK293T-TCR-Lck [15] cells were grown in DMEM (Dulbecco’s Modified Eagle’s Medium) supplemented with 10% FBS and 1% penicillin/streptomycin. The antibiotic puromycin was used to generate and maintain stable cell lines at (0.5 µg/mL) in a cell culture medium. Cells were maintained at 37 °C with 5% CO_2_.

### 2.3. DNA Constructs and Site-Directed Mutagenesis

A PiggyBac Gene Expression Vector pPB[Exp]-Puro-EF1A>hZAP70/T2A/EGFP coding for human Zap70 was generated by VectorBuilder. pEF_hyPBase coding for a hyperactive transposase was purchased from VectorBuilder. The pEYFP-N1-hZap70 vector was also used [16]. Using the Agilent Quick Change II XL (Agilent, Santa Clara, CA, USA) system, site-directed mutagenesis was performed to generate Zap70 mutants according to the manufacturer’s instructions.

The following primers were designed using the tools from Agilent and synthesized by Biomers:Zap70 C564A, forward: tacagttcgggtggagcctctggtgggcactc;reverse: gagtgcccaccagaggctccacccgaactgta.Zap70 C564R, forward: gtgcccaccagagcgtccacccgaact;reverse: agttcgggtggacgctctggtgggcac.Successful mutations were confirmed by sequencing.

### 2.4. Cell Transfections

To generate stable transfectants expressing Zap70^wt^ or Zap70^C564A^, P116 cells were transfected with 5–30 µg of pPB[Exp]-Puro-EF1A>hZAP70/T2A/EGFP and 5 μg of pEF_hyPBase. For transient transfections of P116 cells, we used 5–30 µg of the pEYFP-N1-hZap70 vector. DNA electroporation of P116 cells was performed using the Gene Pulser II System (BIORAD) as previously described [16]. The transfected cells were cultured in an RPMI1640 medium supplemented with 10% FBS and 1% penicillin/streptomycin. A total of 0.5 μg/mL of puromycin (Gibco, Waltham, MA, USA) was added for the generation of stable transfectants, which were additionally sorted with the Aria Cell Sorter 3 (BD Bioscience) and afterwards maintained in RPMI supplemented with 10% FBS, 1% penicillin/streptomycin, and 0.1% Ciprobay.

For the transfection of HEK293T cells, 1 × 10^6^ cells were seeded onto 6-well tissue culture plates one day before transfection. Separately, a mixture was prepared containing 300 μL of 250 mM CaCl_2_ and 10 μg of pEYFP-N1-hZap70, which was added dropwise under constant agitation to 300 μL of HEPES buffered saline (Sigma-Aldrich). After incubation for 45 min at RT, 3 mL of DMEM (PAN Biotech, Aidenbach, Germany) were added to the mixture, which was subsequently carefully pipetted into the well containing the cells. Cells were incubated in the presence of a transfection medium for 24 h.

### 2.5. Stimulation and Lysis of Cells

A total of 2 × 10^6^ P116 T cells were directly stimulated with 10 μg/mL of soluble or plate-bound anti-CD3ε (UCHT1) (BioLegend, San Diego, CA, USA). Subsequently, the cells were lysed in a lysis buffer containing 1% IGEPAL (Sigma Aldrich), 1% n-Dodecyl-β-D-maltoside (LM) (Santa Cruz Biotechnology, Dallas, TX, USA), 50 mM Tris pH 7.5 (Carl Roth GmbH, Karlsruhe, Germany), 165 mM NaCL (Carl Roth GmbH),10 mM NaF (Carl Roth GmbH), 10 mM EDTA (Carl Roth GmbH), 1 mM Natriumorthovanadate (Fluka, Buchs, Switzerland), 1 mM PMSF (Sigma-Aldrich, St. Louis, MI, USA). Samples were assayed using SDS-PAGE and immunoblotting as previously described [16].

### 2.6. Immunoprecipitation

The ζ-chain was immunoprecipitated using agarose-bound anti-CD3ζ (6B.10, Santa Cruz Biotechnology) mixed with BSA 1% and rotated over night at 4 °C. To remove unspecific bindings, the immunoprecipitates were washed 5 times using washing buffer containing 50 mM Tris pH 7.5 (Carl Roth GmbH),165 mM NaCl (Carl Roth GmbH),10 mM NaF (Carl Roth GmbH), 1% Brij™-58 Surfact Amps™ detergent (Thermo Fisher Scientific, Waltham, MA, USA), and 1% LM (Calbiochem, San Diego, CA, USA). The immunoprecipitates were then subjected to SDS-PAGE and immunoblotting as previously described [16].

### 2.7. In Vitro Kinase Assay

A total of 20 × 10^6^ cells were stimulated with 10 μg/mL of anti-CD3ε (UCHT1) (BioLegend) and lysed in 10% Brij™-58 Surfact Amps™ Detergent (Thermo Fisher Scientific), 150 mM NaCl (Carl Roth GmbH), 10 mM NaF (Carl Roth GmbH, Karlsruhe, Germany), 150 mM Tris pH 7.5 (Carl Roth GmbH), 1% PMSF (Sigma-Aldrich), 1 mM Natriumorthovanadate (Fluka), and 1% Protease Inhibitor Cocktail (Sigma-Adrich). Subsequently, Lck was immunoprecipitated for 2 h at 4 °C, using a polyclonal Lck antibody (06-583 MerckMillipore, Burlington, MA, USA), Protein A beads, and BSA 1%. The samples were washed in washing buffer containing 1% Brij™-58 Surfact Amps™ Detergent (Thermo Fisher Scientific), 50 mM Tris pH 7.5 (Carl Roth GmbH), and 165 mM NaCl (Carl Roth GmbH) five times and split in two parts. One part was subjected to SDS-PAGE and immunblotting as an input control, whereas the other part was washed in tyrosine-kinase-buffer containing 40 mM Tris pH 7.5 (Carl Roth GmbH), 20 mM MgCl x6H20 (Carl Roth GmbH), 2 mM MnCl_2_ (Sigma-Adrich), 0.5 mM DTT (Thermo Fisher Scientific), and 0.1% BSA (Sigma-Aldrich). The kinase activity was measured using the ADP-Glo™ Kinase Assay kit from Promega according to the manufacturer’s instructions.

### 2.8. Calcium-Flux

A total of 2 × 10^6^ stable transfected P116 T cells were incubated with Indo-1 (Thermo Fisher Scientific) at 37 °C for 45 min in RPMI 1640 not containing phenol red (Gibco). The cells were then washed with RPMI and incubated in RPMI without phenol red at 37 °C for an additional 45 min. Anti-CD3ε (Biolegend, San Diego, CA, USA) was added to stimulate the cells after 100 s. Ionomycin (10 mg/mL Sigma-Aldrich) was added 1000 s after antibody stimulation as a positive control. Calcium influx was measured using flow cytometry LSR (BD Bioscience). Two emission channels were recorded, BUV395 (379/28 nm) and BUV496 (525/50 nm), over time, and the ratio of BUV395/BUV496 was plotted against time. Kinetic analysis was performed using FlowJo software (BD Bioscience, Franklin Lakes, NJ, USA).

### 2.9. PLA

A total of 0.135 × 10^6^ cells for each condition were starved and seeded onto diagnostic microscope slides (Thermo Fisher Scientific) for 1 h at 37 °C. Cells were either left untreated or treated with 10 µg/mL of anti-CD3ε (OKT3) antibody or 1mM pervanadate (PerV) at 37 °C for 5 min. To fix the cells, 2% PFA was used for 15 min at RT. Next, the cells were permeabilized using 0.5% saponin for 30 min and then blocked. The Duolink kit (Sigma-Aldrich) was used to stain the blocked cells according to the manufacturer’s instructions. Goat anti-CD3ε (1:600, EB12592, Everest Biotech, Bicester, UK) and mouse anti-Lck (1:200, 3A5, Cell signaling, Danvers, MA, USA) were employed. Nuclei were stained with DAPI (Roth). A total of 5–7 images (on average 700 cells) per sample were taken with a confocal microscope (Nikon C2; 60× magnification) and analyzed with the software BlobFinder.

### 2.10. Statistics and Graphics

GraphPad Prism 8 (Dotmatics, San Diego, CA, USA) software was used for the statistical analyses. The statistical significance between groups was determined using an unpaired Welch’s *t*-test, unless otherwise indicated. *p* < 0.05 was the minimum acceptable level of significance. Graphics were also created using GraphPad Prism 8.

## 3. Results

### 3.1. S-Acylation of C564 Is Dispensable for Signal Propagation and T-Cell Activation

A C564R mutation in the C-terminal part of the kinase domain of Zap70 has been found to be associated with severe combined immunodeficiency in an 11-month-old female patient [12]. A recent study has shown that a Zap70^C564R^ mutant is incapable of transmitting the TCR signal leading to the assembly of the LAT signalosome and hence to T-cell activation in the Zap70-deficient P116 T-cell line [13]. In addition to this important finding elucidating the molecular mechanism at the basis of this form of SCID, this work has additionally proposed that C564 is a site of acylation (i.e S-palmitoylation). The authors proposed that the loss of palmitoylation uncouples Zap70 from the phosphorylation of its immediate downstream substrates LAT and SLP76. We thought that the defective function of the C564R mutant could be due to the particular chemical properties of the amino acid substitution rather than to the loss of S-palmitoylation. Indeed, arginine differs from cysteine both in size and in the polarity of the side chain, which is larger in volume and positively charged in arginine. To test whether the defective signal transduction of the C564R mutant is due to the loss of acylation or to the effect of the arginine substitution, we generated a Zap70^C564A^ mutant. A cysteine-to-alanine mutation is the conventional way to study the function of a cysteine’s side chain and its possible involvement in post-translational modifications including sulfenylation, nitrosylation, and importantly for this study, palmitoylation. To investigate the function of Zap70^C564A^, we stably reconstituted the Zap70-deficient P116 Jurkat T-cell variant with either a wild-type Zap70 (Zap70^wt^) or Zap70^C564A^. As a negative control, we also stably reconstituted P116 cells with an empty vector (Zap70^neg^). To avoid the screening of individual clones and to exclude possible adverse effects resulting from insertional mutagenesis, we used bulk stable cell cultures.

To study the signaling function of Zap70^C564A^, stable P116 cultures were stimulated with anti-CD3 antibodies, and cell lysates were assayed by immunoblotting. As shown in Figure 1, Zap70^neg^ P116 T cells displayed a defective inducible phosphorylation of different signaling molecules (Figure 1A) and impaired Ca^2+^ flux (Figure 1B) in response to TCR stimulation. Hence, as expected, Zap70^neg^ P116 T cells were signaling-incompetent. In contrast, P116 T cells stably reconstituted with Zap70^wt^ were signaling-competent, as indicated by the prompt phosphorylation of LAT, PLC-γ1, ERK1/2 (Figure 1A), and by the rapid increase in intracellular Ca^2+^ levels (Figure 1B) upon TCR being triggered. When we analyzed P116 T cells expressing Zap70^C564A^, we surprisingly found that Zap70^C564A^ was fully signaling-competent and capable of inducing the phosphorylation of LAT, PLC-γ1, ERK1/2 (Figure 1A), and Ca^2+^ influx (Figure 1B). Because it has been previously proposed that the Zap70^C564R^ mutant is signaling-incompetent [13], we compared both C564 mutants side by side (Appendix A). In line with the study by Tewari et al. [13], the Zap70^C564R^ was also signaling defective in our hands. In contrast, Zap70^C564A^ was signaling-competent and indistinguishable from the wild type (Appendix A).

The activation of intracellular signaling pathways leads to transcriptional activation. Therefore, we measured the expression of CD69 as a surrogate marker for T-cell activation. As shown in Figure 1C, we did not observe major differences in the expression levels of CD69 between P116 cells expressing WT and Zap70^C564A^ upon TCR stimulation. Collectively, this first set of data indicates that the S-acylation of C564 is not a major regulator of TCR signaling and T-cell activation.

### 3.2. Zap70^C564A^ Regulates Lck Activity and Proximal TCR Signaling

The phosphorylation of Zap70 is regulated by a series of sequentially ordered events, which include (1) the phosphorylation of the ITAMs of the TCR ζ-chain by Lck, (2) Zap70 recruitment to the phosphorylated ITAMs, (3) the Lck-mediated phosphorylation of Zap70 on Y319, and (4) auto- trans-phosphorylation on Y493, leading to full Zap70 activation [2].

Interestingly, the phosphorylation of the Zap70^C564A^ mutant is increased particularly on Y319 (Figure 2A), in agreement with the hyperphosphorylation of Zap70^C564R^ shown by Tewari et al. [13]. Conversely, the phosphorylation of Y493 was not affected by the C564A mutation (Figure 2B). Since the phosphorylation of Y493 is required for the full activation of Zap70 [3], we assume that the overall enzymatic activity of Zap70 is not affected. In agreement with this hypothesis, we did not observe changes in LAT phosphorylation and downstream signaling between cells expressing Zap70^wt^ and Zap70^C564A^ (Figure 1). To investigate the molecular mechanisms responsible for the enhanced phosphorylation of Zap70^C564A^ on Y319, we initially investigated the phosphorylation of the TCR ζ-chain using a phospho-specific antibody. We found that Zap70^C564A^ P116 cells displayed a significantly higher phosphorylation of TCR-ζ on Y142 compared to cells expressing Zap70^wt^ (Figure 2C). To further investigate whether this effect depends on Lck, we took advantage of HEK-293T cells stably expressing a TCR/CD3 complex (hereafter referred to as HEK-TCR) [3], which were additionally modified to stably express Lck (hereafter referred to as HEK-TCR-Lck) [15]. As shown in Figure 2D, the expression of the Zap70^C564A^ in HEK-TCR-Lck cells resulted in a stronger phosphorylation of TCR-ζ compared to cells expressing Zap70^wt^, in line with the data generated in P116 cells. This effect was abrogated when Zap70^C564A^ was expressed in Lck-negative HEK-TCR cells (Figure 2D). Collectively, these data suggest that Zap70^C564A^ regulates the ability of Lck to phosphorylate the ζ-chain of the TCR/CD3 complex.

We next determined whether Zap70^C564A^ regulates the activity of Lck. Therefore, we initially investigated the phosphorylation of the regulatory tyrosines of Lck. We did not find major differences in the phosphorylation of Lck on the activator Y394 between cells expressing Zap70^C564A^ and Zap70^wt^ under a steady state and upon CD3 stimulation (Figure 3A). However, when we analyzed the phosphorylation of the inhibitory Y505, we found that P116 cells expressing Zap70^C564A^ displayed significantly lower levels of Y505 phosphorylation under steady-state conditions (Figure 3B). These data suggest that Lck is more active in the presence of Zap70^C564A^. We have previously shown that the dynamics of Lck activation cannot be accurately monitored by immunoblotting using phospho-specific antibodies [17,18]. Therefore, we directly measured the enzymatic activity of Lck by performing a non-radioactive in vitro kinase assay. Lck was immunoprecipitated from either Zap70^C564A^ or Zap70^wt^ expressing P116 T cells, which were left either unstimulated or stimulated with an anti-CD3 antibody. As a negative control, the Lck^Y394F^ kinse dead mutant was immunoprecipitated from transiently transfected J.Lck cells [14]. The data depicted in Figure 3C clearly show that in cells expressing Zap70^C564A^, Lck displays an enhanced kinase activity under both resting and stimulated conditions. In summary, these data indicate that C564 is a key residue in Zap70 that regulates the kinase activity of Lck and further provide an explanation for the enhanced phosphorylation of the Lck substrates observed in P116 cells expressing Zap70^C564A^.

In order to phosphorylate the ITAMs, Lck must be recruited to the TCR. Recently, it has been shown that a RK motif within CD3ε binds to the SH3 domain of Lck, thus facilitating the localization of Lck to the engaged TCR and promoting the local augmentation of Lck activity [19]. We used proximity ligation assays (PLA) to measure the proximity between Lck and CD3ε in cells expressing Zap70^C564A^ or Zap70^wt^ [19]. To this end, Zap70^C564A^, Zap70^wt^, and Zap70^neg^ P116 T cells were either left unstimulated or stimulated using an anti-CD3ε antibody or pervanadate as positive controls. P116 T cells expressing Zap70^C564A^ displayed an increase in TCR-Lck proximity compared to the cells reconstituted with Zap70^wt^, both under steady-state conditions and upon TCR stimulation, as shown by PLA (Figure 4). This might explain the increased Lck activity and enhanced phosphorylation of Lck substrates in P116 T cells reconstituted with Zap70^C564A^ as described above.

## 4. Discussion

Zap70 acts together with the tyrosine kinase Lck to orchestrate the initiation of TCR signaling. Lck is not only required for the recruitment and the activation of Zap70 at the TCR but also to bring Zap70 in close proximity to its substrate LAT, thus allowing for signal propagation [20]. In addition, Zap70 mediates negative feedback of the TCR pathway by phosphorylating Y192 within the SH2 domain of Lck [21]. It has been proposed that the phosphorylation of Y192 diminishes the interaction between Lck and its positive regulatory phosphatase CD45, thus resulting in the hyper-phosphorylation of Y505 in Lck and in its inactivation [14]. Conversely, we have shown that a phosphomimetic Lck^Y192E^ mutant still retains its enzymatic activity and is able to initiate signaling in thymocytes [15,22], highlighting that the initial steps of TCR activation and signal propagation are not fully understood. In this study, we revealed a new feedback loop involving Lck and Zap70. This feedback appears to be dependent on a critical cysteine residue at position 564 in Zap70. Cells expressing a Zap70^C564A^ mutant display an increased association of Lck with the TCR and increased Lck kinase activity. These observations are consistent with the enhanced phosphorylation of Lck substrates TCR-ζ and Zap70. How Zap70^C564A^ regulates Lck activity is still unclear. It is possible that Zap70^C564A^ directly binds and activates Lck, perhaps via the interaction between the SH2 domain of Lck with phosphorylated Y319 of Zap70. However, immunoprecipitation experiments performed to assess this hypothesis were inconclusive (data not shown). Alternatively, Zap70^C564A^ may favor the interaction of Lck with CD3ε. It has been proposed that the binding of the Lck SH3 domain to an RK motif located in CD3ε results in the local augmentation of Lck activity [19]. The TCR/CD3 complex is in equilibrium between a resting and an open state, the resting state being stabilized by cholesterol binding to the transmembrane regions of the TCR, and the open conformation by the binding of the cognate ligand [23,24]. Only in the open confirmation is the RK motif exposed to allow the binding of Lck to CD3ε [19]. It is possible that Zap70^C564A^ contributes to stabilizing the open conformation of the TCR/CD3 complex, even in the absence of TCR engagement, as suggested by the observation that the kinase activity of Lck and its proximity to the TCR are already increased under steady-state conditions. In agreement with this hypothesis, we found that Zap70^C564A^ is more associated with the TCR under both a steady state and upon CD3 stimulation, although the values were not statistically significant (Appendix A).

Our study additionally shed light on another important proposed mechanism of how Zap70 regulates TCR signaling. A Zap70^C564R^ mutation has been reported to be associated with a form of severe combined immunodeficiency in humans [12]. A detailed characterization of Zap70^C564R^ in Zap70-deficient P116 T cells revealed that this mutant is not capable of propagating TCR signaling for the formation of the LAT signalosome, although it still retains its enzymatic activity [13]. According to the “catch-and-release” model, Zap70 is released from the engaged TCR upon activation, but it is retained at the plasma membrane and translocated into spatially segregated domains containing its substrate LAT [25]. Consequently, Zap70 phosphorylates LAT to propagate the activation signals. However, how Zap70 is kept at the plasma membrane upon release from the TCR remains still unknown. Tewari et al. proposed that Zap70 is S-acylated on C564, and that S-acylation is required to keep Zap70 at the plasma membrane [13]. The proposed model postulates that the Zap70^C564R^ mutant cannot be S-acylated, and hence can no longer localize at the plasma membrane in close proximity to its substrate LAT, resulting in impaired signal propagation. We confirmed that Zap70^C564R^ is indeed signaling defective. In contrast, the Zap70^C564A^ mutant propagates TCR signaling signals despite the fact that, similarly to Zap70^C564R^, it is not S-acylated on C564. Thus, the S-acylation of C564 does not seem to be required for signal propagation. These data further suggest that the block in signal propagation is rather due to the cysteine-to-arginine point mutation, which may have an impact on the ability of Zap70 to interact with LAT. This hypothesis was investigated in a study by Tewari et al., which showed that Zap70^C564R^ is able to phosphorylate its substrate SLP76 in vitro [13]. However, Zap70^C564R^ might still not be able to interact with LAT in an intact cell system. Alternatively, it is possible that Zap70^C564R^ has a higher affinity for the phosphorylated ITAMs than the WT form and hence cannot be released from the TCR, thus impairing the “catch-and-release” mechanism for signal propagation.

During the last few years, investigations on the function of cysteines that are not located within the catalytic center of enzymes, in particular, of protein tyrosine kinases, have significantly grown [26]. Cysteines are targeted by different reversible modifications (e.g., sulfenyation, nitrosylation, and glutathionylation), which may modulate the protein structure and function of different tyrosine kinases including c-Src [27], c-Abl [28], Lyn [29], EGFR [30], and c-Ret [31]. Most recently, we have shown that the sulfenylation of C575 regulates Zap70 stability and activity, possibly by promoting the interaction with the co-chaperone Cdc37 [16]. In line with this finding, we also investigated whether C564 is sulfenylated. We found that the sulfenylation pattern of Zap70^wt^ was comparable to that of Zap70^C564A^ (Appendix A). Thus, we concluded that C564 is not a major target for sulfenylation. Whether C564 is targeted by other post-translational modifications remains to be investigated.

In summary, our study indicates that the S-acylation of C564 is not required for signal propagation. It appears, rather, that C564 is involved in the regulation of a new feedback loop regulating the activity of Lck and proximal TCR signaling. Thus, Zap70 functions are controlled by a number of sophisticated mechanisms involving not only tyrosine resides but also cysteines.

## Figures and Tables

**Figure 1 cells-11-02723-f001:**
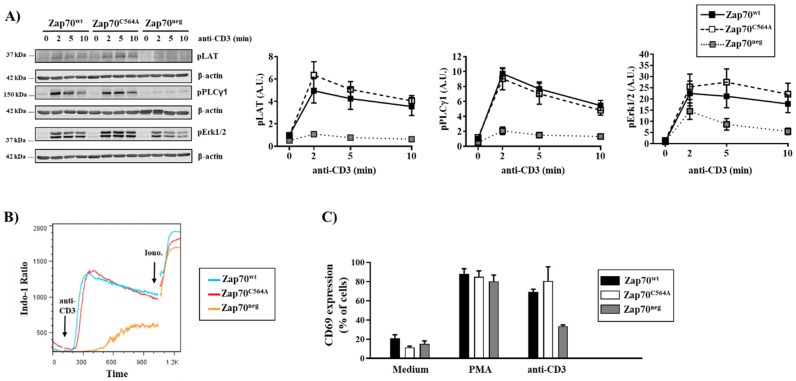
**Signal propagation is intact in T cells expressing Zap70^C564A^.** (**A**) Zap70-deficient P116 T cells stably expressing Zap70^wt^, Zap70^C564A^, or no Zap70 (Zap70^neg^) were stimulated with the anti-CD3 antibody UCHT1 for the indicated time points. Subsequently, the cells were lysed, and the phosphorylation of LAT, PLCγ1, and Erk1/2 was investigated using the indicated phospho-specific antibodies. Equal protein loading was assessed using an anti-β-actin antibody. One representative experiment is shown (*n* = 5). Bands were quantified by normalizing the signal of phosphorylated proteins to the signal of β-actin. Graphs on the right side show the phosphorylation levels of the indicated molecules as arbitrary units ± SEM of five independent experiments. Statistical analyses performed using Welch’s *t*-test revealed no significant differences between samples. (**B**) Calcium influx was assessed by flow cytometry. Ionomycin was used to show the equal loading of the cells with Indo-1. One representative experiment is shown (*n* = 3). (**C**) Transcriptional activation in P116 T-cell clones was assessed 6 h upon stimulation with plate-bound UCHT1 by measuring the level of CD69 (*n* = 3). PMA was used as a positive control. The graph shows the percentage of CD69 expression ± SEM for three independent experiments.

**Figure 2 cells-11-02723-f002:**
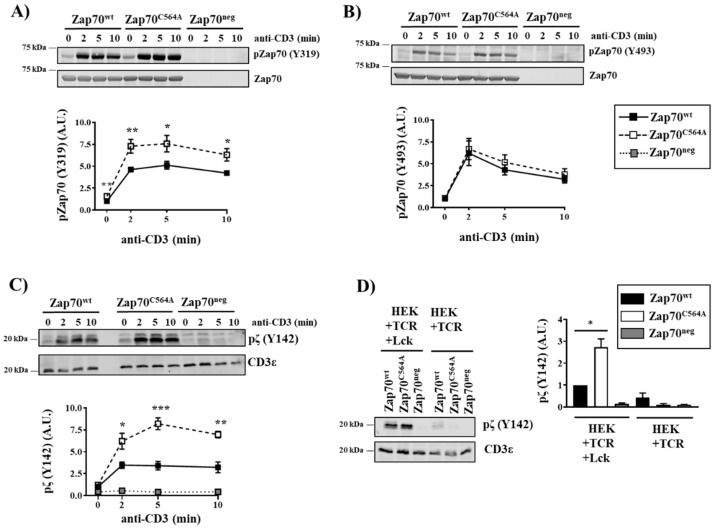
**Enhanced proximal TCR signaling upon the expression of Zap70^C564A^.** Zap70-deficient P116 T cells stably expressing Zap70^wt^, Zap70^C564A^, or no Zap70 (Zap70^neg^) were stimulated with the anti-CD3 (UCHT1) antibody for the indicated time periods. Subsequently, cells were lysed, and the phosphorylation of Zap70 on the regulatory tyrosines Y319 (**A**) and Y493 (**B**), and the phosphorylation of the ζ-chain (Y142) (**C**), were investigated using phospho-specific antibodies. Equal protein loading was assessed using antibodies directed against the total Zap70 or CD3ε. One representative experiment is shown (*n* = 6). Bands in (**A**–**C**) were quantified by normalizing the signal of phosphorylated Zap70 or ζ to total Zap70 or CD3ε, respectively. Graphs in (**A**–**C**) show the phosphorylation levels of the indicated molecules as arbitrary units ± SEM of six independent experiments. (**D**) HEK293T cells stably expressing the TCR or the TCR plus Lck were transiently transfected with plasmids encoding Zap70^wt^, Zap70^C564A^, or no Zap70 (Zap70^neg^). Subsequently, cell lysates were prepared and assayed for the phosphorylation of the ζ-chain by immunoblotting. Immunoblotting using anti-CD3ε was used to show equal loading. Bands in (**D**) were quantified by normalizing the signal of phosphorylated ζ to the CD3ε signal. The graph shows the phosphorylation levels of the ζ-chain as arbitrary units ± SEM of four independent experiments. Significant *p* values were calculated using Welch’s *t*-test (* *p* < 0.05; ** *p* < 0.01; *** *p* < 0.001).

**Figure 3 cells-11-02723-f003:**
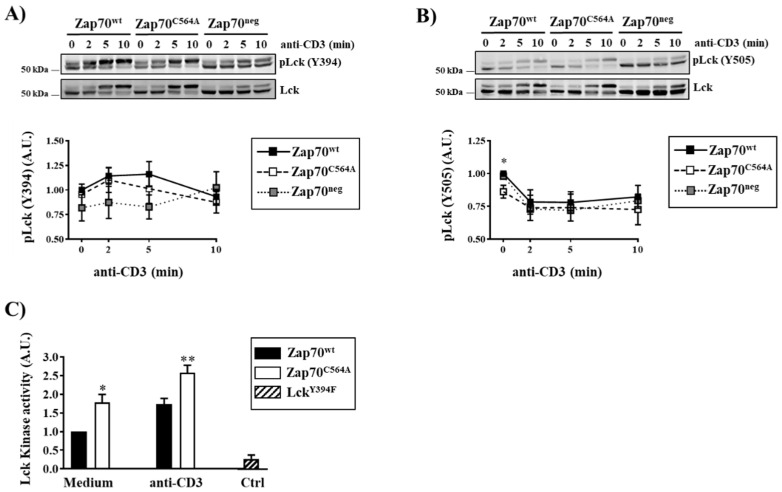
**Enhanced Lck activity in cells expressing Zap70^C564A^.** Zap70-deficient P116 T cells stably expressing Zap70^wt^, Zap70^C564A^, or no Zap70 (Zap70^neg^) were stimulated with the anti-CD3ε UCHT1 antibody for the indicated time periods. Subsequently, cells were lysed, and the phosphorylation of Lck on the regulatory tyrosines Y394 (**A**) and Y505 (**B**) were investigated using phospho-specific antibodies. Equal protein loading was assessed using an antibody against total Lck. One representative experiment is shown (*n* = 6–7). Bands in (**A**,**B**) were quantified by normalizing the signal of phosphorylated Lck to total Lck. Graphs in (**A**,**B**) show the phosphorylation levels of Lck as arbitrary units ± SEM of six to seven independent experiments. (**C**) P116 T cells stably expressing either Zap70^wt^ or Zap70^C564A^ were stimulated with anti-CD3ε (UCHT1) antibodies for 2 min. Subsequently, the cells were lysed, Lck was immunoprecipitated, and its kinase activity was measured using a non-radioactive assay. As a negative control, Lck-deficient J.Lck transiently transfected with the kinase dead mutant Lck^Y394F^ were used. The graph shows the kinase activity of Lck expressed as arbitrary units ± SEM of three independent experiments upon the normalization of the samples to the values of the enzymatic activity of Lck in unstimulated P116 expressing Zap70^wt^, which were set to one. Significant *p* values were calculated using Welch’s *t*-test (* *p* < 0.05; ** *p* < 0.01).

**Figure 4 cells-11-02723-f004:**
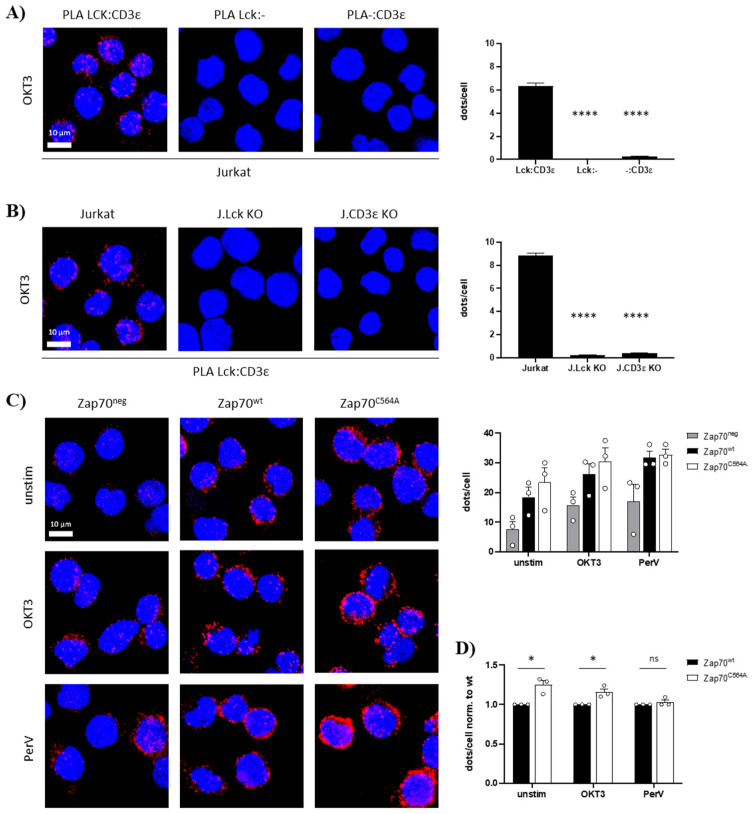
**Expression of Zap70^C564A^ increases proximity between Lck and the TCR.** An in situ Proximity Ligation Assay (PLA) between the TCR (CD3ε) and Lck was performed to study the proximity between Lck and the TCR; a red fluorescent signal indicates that the distance between Lck and the TCR (CD3ε) is less than 80 nm. (**A**) Anti-CD3ε (OKT3) was used to simulate Jurkat cells for 5 min at 37 °C. Technical PLA controls were performed using both anti-Lck and anti-CD3ε primary antibodies (left panel), the anti-Lck primary antibody alone (middle panel), or the anti-CD3ε antibody alone (right panel). In all samples, both secondary antibodies were used. The bar graph shows the quantification of two independent experiments. One-way ANOVA was used to perform statistical analyses. Mean values ± SEM are shown. **** *p <* 0.0001. (**B**) As additional PLA controls, we used parental Jurkat T cells (expressing both CD3ε and Lck) (left panel) and CRISPR/Cas9 gene edited Lck-deficient (J.Lck KO, middle panel) or CD3ε-deficient (J.CD3ε KO, right panel) Jurkat T-cell variants. The PLA was performed as indicated in (**A**). The bar diagram shows the quantification of two independent experiments. Statistical analysis was performed as in (**A**). (**C**) Zap70-deficient P116 cells stably expressing no Zap70 (Zap70^neg^), Zap70^wt^ or Zap70^C564A^ were either left unstimulated, stimulated with anti-CD3ε (OKT3), or stimulated with pervanadate (PerV) for 5 min at 37 °C. Nuclei were stained with DAPI. Data of three independent experiments are shown (upper right bar diagram), *n* = ~700 cells per sample. (**D**) Values were set to one for ZAP70^wt^ for each experiment and each condition to best visualize the effect of the C564A mutation. Statistical analysis was performed using unpaired Student’s *t* test. Mean values + SEM are shown, ns = not significant; * *p <* 0.1.

**Table 1 cells-11-02723-t001:** FACS antibodies.

Antibody	Clone	Company
APC anti-human CD3	HIT3a	BD Pharmingen™
APC anti-human CD69	FN 50	BD Pharmingen™
Goat IgG anti-Mouse IgG + IgM (H + L)	polyclonal	Dianova

**Table 2 cells-11-02723-t002:** Antibodies for western blot, immunoprecipitation, and PLA.

Antibody	Clone/Lot	Company
β-actin	AC-15	Sigma-Aldrich
CD3ε	EB12592	Everest Biotech Ltd.
Lat pY191	3584S	Cell Signaling Technology
Lck	06-583	Merck Millipore, upstate
Lck	3A5	Santa Cruz Biotechnology
Lck pY505	2751S	Cell Signaling Technology
PLCγ pY783	2821S	Cell Signaling Technology
p44/42 MAPK (Erk1/2) (pT202/pY204)	D13.14.4E	Cell Signaling Technology
Src pY416	2101S	Cell Signaling Technology
Zap70 pY319	2701S	Cell Signaling Technology
Zap70 pY493	2704S	Cell Signaling Technology
Zap70	1E7.2	Santa Cruz Biotechnology
CD3 ζ pY142	SAB4301233	Sigma-Aldrich
CD3 ζ	6B10.2	Santa Cruz Biotechnology

**Table 3 cells-11-02723-t003:** Antibodies for T-cell stimulation.

Antibody	Clone	Company
CD3ε	UCHT1/OKT3	BioLegend

## Data Availability

The data presented in this study are available on request from the corresponding authors.

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
