# Peer review of "A Cysteine Residue within the Kinase Domain of Zap70 Regulates Lck Activity and Proximal TCR Signaling"

_cells, 2022, doi:10.3390/cells11172723_

Round 1

Reviewer 1 Report

Previously, a non-palmitoylable C564R Zap70 mutant was described in a patient suffering from mmunodeficiency. Here, Schultz et al engineered a C564A mutant of Zap70, which similarly to Zap70C564R is non-palmitoylatable, but was capable of propagating TCR signaling. In various in vitro models they revealed that Zap70C564A shows enhanced activity of Lck, increased its proximity to the TCR and ensured hyperphosphorylation of
TCR-ζ and Zap70 at Y319. Thus the authors convincingly demonstrated, that C564 is important for the regulation of Lck activity and proximal TCR signaling, but not for the palmitoylation of Zap70 as suggested by Tewari et al.

Overall, the paper is clearly written and the message is convincing.The experiments are done thoroughly and therefore the study should be published, if some minor points are addressed:

1. Figure 2: Please indicate in line 262 (Fig. 2A) and shortly state that there is no difference found on the phosphorylation on Y493 (Fig. 2B). In line 268, this should be 2C instead of 2B and in line 271 2D instead of 2C.

2. Figure 3: please move the Figure to line 295, then the figure can be directly seen during reading the text above.

3. Figure 4: please label the cells in part C as done in A and B. Then it's more clear which cells are shown. The normalization is not clear to me. Did the authors always set the WT of each experiment to 1. I would suggest to set the mean of WT to 1 and the calculate the ratio. Additionally, it should be stated how many cells were counted in each experiment.

4. It would be nice to show the sulfenylation pattern (mentioned in line 407) as a supplement.

5. In my opinion there is a wrong wording in line 414: it should be "only" instead of "also", or?

Author Response

Reviewer1.

  1. Figure 2: Please indicate in line 262 (Fig. 2A) and shortly state that there is no difference found on the phosphorylation on Y493 (Fig. 2B). We are thankful to the Reviewer for this suggestion. We have now added a sentence stating that the phosphorylation of Y493 is not changed.

In line 268, this should be 2C instead of 2B and in line 271 2D instead of 2C. We apologize with the reviewer for this mistake, which has been now corrected.

  1. Figure 3: please move the Figure to line 295, then the figure can be directly seen during reading the text above. According to Reviewer´s suggestion, figure 3 has been now moved. Nevertheless, because of the several changes in the formatting, it is difficult to place Fig. 3 on the same page with the corresponding text.

  1. Figure 4: please label the cells in part C as done in A and B. Then it's more clear which cells are shown. The normalization is not clear to me. Did the authors always set the WT of each experiment to 1. I would suggest to set the mean of WT to 1 and the calculate the ratio. Additionally, it should be stated how many cells were counted in each experiment. We thank the Reviewer for the suggestions. Now, we have labeled the cells in C as done in A and B to improve clarity. We have specified in the figure legend how the normalization of panel D was done. Indeed the number of dots/cell of the WT was set to one for each experiment and the ratio between the WT and the mutant calculated to best visualize the effect of the C546A mutation. By setting the mean of WT to 1 and then calculating the ration the results were unchanged. We have also clarified in the methods section and in the figure legend that 5-7 images were quantified per sample, with a average of ~700 cells quantified per sample.
  2. It would be nice to show the sulfenylation pattern (mentioned in line 407) as a supplement. According to Reviewer´s suggestion, we have now included these data in supplementary figure 3.

  1. In my opinion there is a wrong wording in line 414: it should be "only" instead of "also", or? Unfortunately, we didn´t find the wrong wording in line 414.

Reviewer 2 Report

A non-palmitoylatable C564R ZAP-70 mutant, which has been associated to a form of SCID, has been previously reported to inhibit TCR signaling, suggesting a role for S-acylation of ZAP-70 in TCR coupling to the LAT signalosome that promotes T cell activation. Here the authors have hypothesized that the effects of the C564R substitution may be caused not by lack of palmitoylation but by the specific properties of the arginine replacing the cysteine. To test this hypothesis they investigated the outcome of C564 substitution with alanine. They show that, at variance with C564R ZAP-70, C564A ZAP-70 does not impair TCR signaling. Of note, C564A ZAP-70 expression results in enhanced ZAP-70 phosphorylation, whis is associated to enhanced CD3zeta phosphorylation. This is in turn caused by enhanced Lck activity and Lck proximity to CD3epsilon, an event that faciltates Lck activation. The authors conclude that S-acylation is not required for TCR signaling but is implicated in a feedback loop where ZAP-70 limits Lck activity.

The results are novel and interesting, and the manuscript is clearly written and logically articulated. However, a major question is why, if Lck and ZAP-70 are hyperactive in the presence of the C564A mutation, TCR signaling downstream of ZAP-70 is not enhanced as well (as shown in figure 1). Also, although the authors address potential mechanisms in the Discussion, it would be interesting to test some of these mechanisms.

Other points

Point 1. Figure 1A. Immunoblots should be quantified over multiple experiments and plotted as histograms (with stats). Also, significant Erk phosphorylation can be detected in the ZAP-70 negative cells. The authors should comment on this.

Point 2. Figure 3B. The blot shown does not appear representative of the quantification, which indicates an enhanced Y505 Lck phosphorylation under steady-state conditions.

Point 3. Figure 4C. Please indicate the ZAP-70 mutants for each column of panels. Also, note that the top left panel (wt ZAP-70, unstimulated) shows an image under the one shown.

Point 4. Please indicate the migration of molecular weight markers for all immunoblots.

Point 5. In the Discussion (lines 369-371) the authors suggest the possibility that C564A may contribute to stabilize the open conformation of the TCR/CD3 complex, even in the absence of TCR engagement". Does C564A ZAP-70 interact with the TCR in the absence of stimulation? Similarly, they discuss a possible mechanisms involving the effect of the C564 mutations on tthe ability of ZAP-70 to interact with LAT. Have they checked whether C564A ZAP-70 interacts with LAT upon TCR stimulation?

Point 6. In the Discussion (lines 410-412) the authors state that " S-acylation of C564 is not required for the assembly of the LAT signalosome". Although the data suggest that this may be the case, the authors have not looked at the LAT signalosome and hence the statement should be modified.

Author Response

Reviewer2.

However, a major question is why, if Lck and ZAP-70 are hyperactive in the presence of the C564A mutation, TCR signaling downstream of ZAP-70 is not enhanced as well (as shown in figure 1). We thank the reviewer for this important comment. We did not state that Zap70 is hyperactive in the manuscript. We observed that Zap70 is hyperphosphorylated on Y319. Conversely, phosphorylation of Y493 in the kinase domain, which is required for the full activation of Zap70, is comparable between WT and C564A Zap70. Therefore, we assume that the overall activity of Zap70 is not changed and hence, we do not detect differences in downstream signaling. We have changed the text in the manuscript to make this point clearer.    

Point 1. Figure 1A. Immunoblots should be quantified over multiple experiments and plotted as histograms (with stats). As suggested from the Reviewer, we have added graphs from multiple experiments. The differences between samples were not statistically significant.

Also, significant Erk phosphorylation can be detected in the ZAP-70 negative cells. The authors should comment on this. It has been previously demonstrated that the Raf-MEK-ERK1/2 pathway can be activated in P116 cells, likely via a Fyn-PLCγ1-DAG-Ras pathway (Shan et al., Mol Cell Biol 2001; PMID: 11585897). As described in the work from Shan and co-workers and as shown in Figure 1A, Erk activation in P116 cells has a transient rather than a sustained kinetic.

Point 2. Figure 3B. The blot shown does not appear representative of the quantification, which indicates an enhanced Y505 Lck phosphorylation under steady-state conditions. We apologize with the Reviewer for this point. However, the difference in pY505 between WT and C564A Zap70 is just 15% and hence, not clearly visible on WB. We have additionally replaced the previous blots with new ones. We hope now that the difference is clearer.

Point 3. Figure 4C. Please indicate the ZAP-70 mutants for each column of panels. Also, note that the top left panel (wt ZAP-70, unstimulated) shows an image under the one shown. We thank the Reviewer for the suggestions. Now, we have labeled the panels in C as suggested to improve clarity. We have taken care of the quality of the figure as requested.

Point 4. Please indicate the migration of molecular weight markers for all immunoblots. As suggested by the reviewer, we have now indicated the size of the molecular weight markers. For beta-actin, we have indicated its real molecular weight because the bands of the molecular weight markers ( 37 kDa and 50kDa) are outside of the blot.

Point 5. In the Discussion (lines 369-371) the authors suggest the possibility that C564A may contribute to stabilize the open conformation of the TCR/CD3 complex, even in the absence of TCR engagement". Does C564A ZAP-70 interact with the TCR in the absence of stimulation? We thank the reviewer for this point. We indeed checked the association between Zap70 and the TCR by performing ζ IPs. We found that Zap70 C564A interacts more with the TCR both in steady state as well as upon CD3 stimulation. However, the difference between Zap70 wt and C564A was not statistically significant. We have now included a graph with the data in a supplementary figure (Fig. S2) and changed the text in the discussion accordingly.

Similarly, they discuss a possible mechanisms involving the effect of the C564 mutations on the ability of ZAP-70 to interact with LAT. Have they checked whether C564A ZAP-70 interacts with LAT upon TCR stimulation? In the discussion, we discussed about the possibility that C564R does not interact with LAT. As we do not see any difference in LAT phosphorylation in cells expressing Zap70 C564A, we assume that Zap70 C564A normally interacts with its substrate. Therefore, we did not perform this experiment.

Point 6. In the Discussion (lines 410-412) the authors state that " S-acylation of C564 is not required for the assembly of the LAT signalosome". Although the data suggest that this may be the case, the authors have not looked at the LAT signalosome and hence the statement should be modified. We have removed this statement according to the Reviewer´s suggestion.

Reviewer 3 Report

Schultz et al. report a new regulatory mechanism governing TCR signaling cascade involving the ZAP-70 kinase. Previous works had shown that a mutation to arginine of a cysteine residue of ZAP-70 (Cys-564) blocks TCR signaling without affecting its kinase activity, and it was proposed that palmitoylation of that cysteine residue was needed for the localization of ZAP-70 close to the plasma membrane and the generation of downstream signals. In their manuscript, Schulz et al. generate a C564A mutant of ZAP-70 which, like the C564R mutant, is not palmitoylable, but is capable of transducing downstream signals. Moreover, cells expressing the C564A mutant seem to show enhanced activity of Lck, probably due to an increased proximity to the TCR.

The present work makes a notable contribution to the field of TCR signaling and the corresponding negative regulation. The manuscript is well written and easy to read. The technology used is appropriate, and the group has extensive experience in the field. However, some points should be addressed:

The main weakness of the work is the fact that it does not make a comprehensive comparison with the C564R mutant. Although the supplementary figure presented shows that the C564R mutant is not able to induce signals leading to PLC-gamma activation (the image of LAT-Y191 phosphorylation is not good), no comparative Lck-CD3 PLA analysis is made in cells expressing both ZAP-70 mutants. This would be of interest as a proof that this mutation affects the proximity of these two molecules, and would help to explain the very different effects caused by two different mutations of the same residue.

Minor points to be addressed:

- Page 7, line 268, change "Fig. 2B" to "Fig. 2C".

- Page 7, line 271, change "Fig. 2C" to "Fig. 2D".

- In page 8, Fig. 3B does not show a clear difference in Lck-Y-505 basal phosphorylation between cells expressing ZAP-70-wt and the ZAP-70-C564A mutant. In addition, the bar graph shows no difference in the basal phosphorylation of Lck between ZAP-70-neg and ZAP-70-wt expressing cells, but the WB image suggests that the absence of ZAP-70 increases Lck-Y-505 phosphorylation. The authors should clarify this point.

- In page 8, Fig. 3B the authors compare the kinase activity of Lck from cells expressing ZAP-70-wt or the ZAP-70-C564A mutant, showing that the mutant form of ZAP-70 increases the kinase activity of Lck. It would be of interest to compare whether the absence of ZAP-70 (in P116 cells) is similar to the one in the parental Jurkat cell line or in P116 expressing the wild-type form of ZAP-70. This would be of interest to reinforce the role of ZAP-70 in a negative feedback loop.

- Page 9. Figure 4C should indicate the different cell types in the images for ease of understanding by the readers.

Author Response

Reviewer 3.

The main weakness of the work is the fact that it does not make a comprehensive comparison with the C564R mutant. Although the supplementary figure presented shows that the C564R mutant is not able to induce signals leading to PLC-gamma activation (the image of LAT-Y191 phosphorylation is not good), no comparative Lck-CD3 PLA analysis is made in cells expressing both ZAP-70 mutants. This would be of interest as a proof that this mutation affects the proximity of these two molecules, and would help to explain the very different effects caused by two different mutations of the same residue. We thank the Reviewer for this comment. The aim of this study was to characterize the signaling function of the C564A Zap70 mutant, which is signaling competent. Based on the data published by Tewari et al., we were surprised by our finding. We thought that this discrepancy was due to different P116 clones or to other experimental conditions used by us and by Tewari and co-workers. Therefore, we decided to assess whether also under our experimental conditions the C564R mutant was signaling defective. We agree that a comprehensive comparison between C564A and C564R is of interest. We are not sure that both the C-to-R and C-to-A mutations affect proximal signaling in the same why. To investigate this point, many additional experiments are required and we fear that this is beyond the aim of this study.

We have also replaced the pLat blot in supplementary figure 1, as suggested by the Reviewer.      

- Page 7, line 268, change "Fig. 2B" to "Fig. 2C". We apologize with the Reviewer for this mistake, which has been now corrected.

- Page 7, line 271, change "Fig. 2C" to "Fig. 2D". We have also corrected this mistake.

- In page 8, Fig. 3B does not show a clear difference in Lck-Y-505 basal phosphorylation between cells expressing ZAP-70-wt and the ZAP-70-C564A mutant. This point was also raised by Reviewer2. We apologize with the Reviewer for this point. However, the difference in pY505 between WT and C564A Zap70 is just 15% and hence, not clearly visible on WB. We have additionally replaced the previous blots with new ones. We hope now that the difference is clearer.

In addition, the bar graph shows no difference in the basal phosphorylation of Lck between ZAP-70-neg and ZAP-70-wt expressing cells, but the WB image suggests that the absence of ZAP-70 increases Lck-Y-505 phosphorylation. The authors should clarify this point. We agree with the reviewer that phosphorylation of Y505 seems to be stronger in Zap70-deficient cells. However, we would like to point out that Zap70-deficient cells also reproducibly express more Lck (about 30% more). When the signal of pY505 was normalized to the signal of total Lck, we do not see any difference between P116 transfected with the empty vector and reconstituted with Zap70 WT (Please see graph in Fig.3B) 

- In page 8, Fig. 3B the authors compare the kinase activity of Lck from cells expressing ZAP-70-wt or the ZAP-70-C564A mutant, showing that the mutant form of ZAP-70 increases the kinase activity of Lck. It would be of interest to compare whether the absence of ZAP-70 (in P116 cells) is similar to the one in the parental Jurkat cell line or in P116 expressing the wild-type form of ZAP-70. This would be of interest to reinforce the role of ZAP-70 in a negative feedback loop. We thank the Reviewer for raising this important point. We did not include P116 cells in our experiments because we feared that the regulation of Lck activity in Zap70-deficient cells is a bit out of our aim. We also fear that the phenomenon is even more complicated from what the Reviewer has proposed. Indeed, the analysis of the phosphorylation of Lck regulatory sites shown in Fig.3 indicates that in the absence of Zap70 the phosphorylation of the activatory Y394 is reduced in Zap70-deficient cells (Fig. 3A), whereas the phosphorylation of the negative regulatory site Y505 (Fig. 3B) is comparable. Therefore, it is likely that the kinase activity of Lck is reduced in the absence of Zap70 and not increased as proposed by the Reviewer. Despite the fact that we agree with the reviewer that issue is important but in our opinion it requires additional experiments and a separate study.

- Page 9. Figure 4C should indicate the different cell types in the images for ease of understanding by the readers. We apologize with the Reviewer for the lack of clarity. This mistake has been now corrected.

Round 2

Reviewer 2 Report

The authors have addressed satisfactorily the issues raised in my previous review.